# Room-Temperature Cholesteric Liquid Crystals of Cellulose Derivatives with Visible Reflection

**DOI:** 10.3390/polym15010168

**Published:** 2022-12-29

**Authors:** Yuki Ogiwara, Tatsuya Suzuki, Naoto Iwata, Seiichi Furumi

**Affiliations:** Department of Chemistry, Graduate School of Science, Tokyo University of Science, 1-3 Kagurazaka, Shinjuku, Tokyo 162-8601, Japan

**Keywords:** cellulose, cholesteric liquid crystal, thermotropic, side chain, visible reflection, room temperature

## Abstract

Hydroxypropyl cellulose (HPC) derivatives with alkanoyl side chains have attracted attention as bio-based cholesteric liquid crystal (CLC) materials with reflection colors. By taking advantage of the ability to change the reflection color in response to external stimuli, the thermotropic CLCs can be applied to a wide variety of photonic devices for a sustainable society of future generations. However, the thermotropic CLCs of HPC derivatives substituted with only one kind of alkanoyl group are not suitable for such applications because they do not exhibit visible reflection at room temperature. In this report, we describe a promising strategy to control the reflection colors of HPC derivatives at room temperature by introducing two kinds of alkanoyl groups with different lengths into the side chains of HPCs, which also enables the fine control of temperature dependence on the reflection wavelength. By chemically optimizing the side chain, we successfully prepared room-temperature thermotropic CLCs of HPC derivatives with visible reflection. This report would contribute toward the development of versatile photonic applications by CLCs produced from biomass.

## 1. Introduction

Cellulose is a polysaccharide that consists of glucose as a repeating unit. Recently, cellulose and its derivatives are gaining attention as a biomass resource in today’s world, where the realization of a sustainable society is a pressing issue [1]. This is because cellulose is one of the most abundant biopolymers on the Earth, and might not become depleted, unlike petroleum resources. Cellulose and its derivatives are widely used in many industries, not only as environment-friendly materials, but also as functional materials with excellent mechanical strength or liquid crystallinity [1,2,3]. For example, cellulose nanocrystals are often used as a filler to enhance the mechanical strength of resins as robust materials [4,5,6]. On the other hand, its aqueous dispersion exhibits liquid crystalline phases arising from its rod-like structure as a soft material [7,8].

Among the various liquid crystalline phases, some derivatives of cellulose can exhibit a cholesteric liquid crystal (CLC) phase, which is well known for having unique physical properties of light reflection [7,8,9,10,11,12,13,14,15]. Such CLC structures are commonly found in insects and other organisms, often leading to the emergence of metallic and brilliant reflection colors [16,17]. These colors of CLCs are derived from Bragg reflection. In CLCs, the layers, in which rod-shaped liquid crystalline molecules are oriented, accumulate with slight changes in the azimuthal orientation vectors to form a helical molecular structure. The rod-like molecules of CLCs exhibit anisotropy in the refractive index. Therefore, the helical molecular structure of CLCs brings about the periodic modulation of the refractive index, thereby resulting in the emergence of photonic band gap(s) as photonic crystals in the photon dispersion diagram, due to anomalous group velocity near the edge of the Brillouin zone [18]. The reflection peak wavelength (*λ*) of CLCs, corresponding to the region of the photonic band gap, can be determined by the following equation suggested by de Vries [19]:(1)$$\lambda =n_{\mathrm{av}}p$$
where *n*_av_ and *p* represent the average refractive index and helical pitch length of the CLC, respectively. Here, *p* means the geometric distance between layers until the orientation vectors rotate 360 degrees. Therefore, the *p* value is determined by a distance between the adjacent layers (*d*) and a helical twisting angle (*φ*) of the CLC helical structure, according to the following equation:(2)p=360d/φ

The reflection peak at *λ* can be shifted by controlling the *p* value, which is related with CLC structural parameters of *d* and *φ*, according to Equation (2). The *p* value can be changed in response to various external stimuli, such as the addition of solvent [9,20], temperature [13,21], and so forth [22,23]. As a result of their unique properties, CLC materials have potential applications to a wide variety of photonic devices, such as mechanochromic sensors [24,25], temperature sensors [26,27,28], and wavelength-tunable lasers [26,29,30]. 

Hydroxypropyl cellulose (HPC), which is one of the derivatives of cellulose, is well known to exhibit a thermotropic CLC phase by chemically modifying the terminal hydroxy groups of HPC [12,13,14]. Cellulose nanocrystals are also known to possess the intrinsic capability to generate the CLC phase. However, they can exhibit a CLC phase only when they are dispersed in water at appropriate concentrations [8,11], corresponding to a sort of lyotropic CLC phase. Thus, thermotropic CLCs of HPC derivatives are easy to handle, rather than the lyotropic CLCs of cellulose nanocrystals. This is because the reflection properties of lyotropic CLCs of cellulose nanocrystals are quite vulnerable under dry atmospheric conditions due to concentration increase by natural evaporation of water as the dispersion medium. On the other hand, some kinds of HPC derivatives show sufficient stability in the thermotropic CLC phase, with visible reflection even under dry conditions which arises from the solvent-free system of thermotropic CLC materials. Another technological advantage is that the chemical modifications of HPC allow for on-demand control of the reflection properties of thermotropic CLCs.

In this context, numerous studies have been reported on the thermotropic and lyotropic CLC materials of HPC derivatives from an optical perspective. For example, Watanabe and co-workers investigated the side chain effect of esterified HPC derivatives, with solely one kind of an alkanoyl group on the reflection properties of thermotropic CLCs. The reflection peak wavelength of HPC derivatives can be controlled by the side chain length, that is, the length of the alkanoyl group [21]. According to their report, the reflection peak tends to red-shift by increasing the side chain length. In other words, the thermally induced wavelength shift of the reflection peak is also crucially dependent on the side chain length. Moreover, the heating treatment of esterified HPC derivatives over ~90 °C is requisite for the emergence of a reflection peak of thermotropic CLCs in a full visible wavelength range, by sweeping the temperature. As another strategy, the addition of solvent in HPC derivatives results in easy preparation of CLCs with visible reflection characteristics at room temperature, which are called lyotropic CLCs [20,31]. Such lyotropic CLCs are cumbersome to handle because their reflection properties are easily changed by unintended volatilization of the solvents in lyotropic CLCs. Taking this perspective in account, thermotropic CLCs of cellulose derivatives with visible reflection at room temperature are very attractive soft materials for sustainable photonic device applications.

In this study, we succeeded in the preparation of a series of esterified HPC derivatives as thermotropic CLCs with blue, green, or red reflection features, even at room temperature. For this objective, we designed HPC derivatives possessing two kinds of alkanoyl groups, with various chain lengths in their side chains. The reflection properties of HPC derivatives with alkanoyl groups of 3–18 carbons were investigated in detail. To the best of our knowledge, there are no reported cases of reflection properties of HPC derivatives with such long-chain alkanoyl groups [21,32,33]. As a result, an interesting phenomenon was observed: as the length of the side chain increased, there was a tendency for changes in reflection properties. The chemical optimization of the side chains in esterified HPC derivatives enabled the preparation of room-temperature thermotropic CLCs with visible reflection. This study could contribute to the development of photonic devices utilizing cellulose for the realization of a sustainable society.

## 2. Experimental Section

We synthesized a series of HPC derivatives that possessed two different kinds of alkanoyl side chains through the esterification of all hydroxy groups in HPC with linear alkanoyl chlorides. The introduction ratio of two kinds of alkanoyl groups was controlled to be approximately 9:1 in a stoichiometric manner. Thereafter, we defined the HPC derivatives substituted with two different alkanoyl groups as HPC-X/C*n*. The X means the abbreviation of the main alkanoyl group. In this study, X is -Pr, -Bu, or -Pe, where they refer to propionyl, butyryl, or pentanoyl groups, respectively. The C*n* stands for the abbreviation of another alkanoyl group, where the value of *n* is the number of carbon atoms excluding that at position one. Note that the alkanoyl groups of Pr, Bu, and Pe are the same abbreviations as C2, C3, and C4, respectively. These three alkanoyl groups were denoted in the two ways, in order to make them easier to distinguish between the main alkanoyl group and the other alkanoyl group. For comparison, HPC derivatives modified by a single alkanoyl group, defined as HPC-X (X = -Pr, -Bu, and -Pe), were also synthesized. The typical synthesis procedure of HPC-Pr/C*n* as an example is represented in Figure 1.

As a starting material, we used HPC (viscosity of 2.0 wt% aqueous solution: 2.0–2.9 mPa·s at 20 °C) of FUJIFILM Wako Pure Chemical Co. (Osaka, Japan). The weight average molecular weight (*M*_w_) and number average molecular weight (*M*_n_) of HPC were determined to be 4.45 × 10^4^ and 2.30 × 10^4^, respectively, by measurement with size-exclusion chromatography (HLC-8220 GPC, TOSOH, Tokyo, Japan), using tetrahydrofuran as the eluent and polystyrene standards as the calibration. The average number of hydroxypropyl groups per anhydroglucose unit, that is, the molar substitution (*MS*) value, was determined to be 3.97 by measuring the ^1^H-NMR spectrum of pristine HPC in CDCl_3_ according to a method in a previous report [34]. Therefore, the molecular weight of the anhydroglucose monomer unit of HPC was calculated from the *MS* value to be 392 on average. Super dehydrated acetone and anhydrous pyridine were used as the reaction solvent and base catalyst, respectively, and were also purchased from the FUJIFILM Wako Pure Chemical Co. Alkanoyl chlorides (C*_n_*H_2*n*+1_COCl; *n* = 2, 3, 4, 5, 6, 7, 9, 11, 13, 15, and 17) and poly(vinyl alcohol) (PVA, *M*_w_ = 1.3–2.3 × 10^4^, 87–89% hydrolyzed degree) were purchased from Tokyo Chemical Industry Co., Ltd. (Tokyo, Japan) and Sigma-Aldrich (St. Louis, MO, USA), respectively. These reagents were used as received.

We synthesized 27 kinds of esterified HPC derivatives: HPC-Pr, HPC-Pr/C*n* (*n* = 3, 4, 5, 6, 7, 9, 11, 13, 15, and 17), HPC-Bu, HPC-Bu/C*n* (*n* = 2, 4, 5, 6, 7, 9, 11, 13, 15, and 17), HPC-Pe, and HPC-Pe/C*n* (*n* = 2, 3, 5, and 6). The synthesis procedures were determined with reference to our previous study [35]. The amounts of alkanoyl chlorides used in each synthesis are listed in Table 1 for HPC-Pr/C*n*, Table 2 for HPC-Bu/C*n*, and Table 3 for HPC-Pe/C*n*. Here, the synthesis of HPC-Pr/C3 is described as an example. 

Before the synthesis, pristine HPC was dried under vacuum at room temperature for over 12 h in advance. In a 200 mL three-neck round-bottom flask filled with nitrogen, pristine HPC (2.00 g, 1.00 eq. to the number of anhydroglucose units in 2.00 g of HPC) was completely dissolved in anhydrous acetone (30.0 mL). Subsequently, anhydrous pyridine (2.42 g, 0.98 g/mL, 30.6 mmol, 6.00 eq. to the number of anhydroglucose units in 2.00 g of HPC) was added into the HPC solution. After that, butyryl chloride (C_3_H_7_COCl; 0.16 g, 1.02 g/mL 1.53 mmol, 0.30 eq. to the number of anhydroglucose units in 2.00 g of HPC) was portion-wise added into the reaction solution at room temperature. Successively, the esterification proceeded with heating at 55 °C under reflux for 4 h. As esterification occurred with remaining hydroxy groups of HPC and another alkanoyl group, propionyl chloride (C_2_H_5_COCl; 2.41 g, 1.06 g/mL, 26.0 mmol, 5.10 eq. to the number of anhydroglucose units in 2.00 g of HPC) was added dropwise into the reaction solution under continuous stirring and heating at the same temperature. After reacting for an additional 20 h, the reaction solution was purified by dropping the solution into 500 mL of ultrapure water. The precipitated white, sticky product was completely dissolved in a small amount of acetone, and then the solution was dropped into ultrapure water again. This purification process was repeated four times. Finally, the product was obtained after drying in an oven at 60 °C to remove the solvents. All HPC derivatives synthesized in this study were obtained as sticky polymer melts. These HPC derivatives exhibited a thermotropic CLC phase in the absence of any additives, such as solvents or chiral dopants, as will be discussed later.

The numbers of hydroxy groups substituted with C*_n_*H_2*n*+1_COCl per anhydroglucose unit, denoted as *XE* or *CnE*, were determined by measuring ^1^H-NMR spectra in CDCl_3_ (Ultrashield 400 Plus spectrometer, Bruker, Billerica, MA, USA). The esterification was also confirmed from an FT-IR spectroscope (FT-IR4700, JASCO, Tokyo, Japan) equipped with an attenuated total reflection (ATR) unit with a diamond prism (ATR PRO ONE, JASCO). The degradation of the HPC derivatives during the heating process was monitored using a thermogravimetric analysis (TGA) system (TGA 2010SA, NETZSCH, Selb, Germany). The TGA measurements of pristine HPC and HPC-Pr were conducted between 25 °C and 350 °C, at a heating rate of 10 °C/min under a nitrogen atmosphere.

Reflection properties of the HPC derivatives were investigated by measuring transmission spectra upon changing the temperature. To fabricate the CLC cell, an aqueous solution of PVA with 2.0 wt% was prepared on a glass substrate by spin-coating it at 800 rpm for 10 s, and then at 2000 rpm for 20 s using a spin-coater (ACT-220D II, Active, Saitama, Japan). After drying the glass substrates coated with a PVA thin film, their surface was uniaxially rubbed 50 times with a stick wrapped in a cupra. The HPC derivatives as sticky polymer melts were sandwiched between a pair of the substrates with a gap of ~200 μm on a hot stage at 100–120 °C. Then, the HPC derivatives were applied by shear orientation treatment, which was conducted by manually sliding the upper substrate along the uniaxially rubbing direction, in order to prepare a CLC alignment state with a relatively clear reflection peak. Importantly, the HPC derivatives were not solid-state films or powders but polymer melts. The detailed results are explained in Section 3.1. Transmission spectra of the CLC cells were acquired using a compact charge-coupled device (CCD) spectrometer (USB2000+, Ocean Optics, Orlando, FL, USA) and a tungsten halogen white light source (HL2000, Ocean Optics). The temperature of the CLC cell was precisely controlled using a temperature controller (HS1, Mettler-Toledo, Columbus, OH, USA) equipped with a hot stage (HS82, Mettler-Toledo). Reflection spectra of the CLC cells were recorded using the above-mentioned spectrometer and halogen light source. An optical fiber with a reflection probe was used for the measurements. All reflection spectra in this study were measured at room temperature of ~25 °C.

## 3. Results and Discussion

### 3.1. Synthesis and Characterization of HPC-X/Cn

The chemical structures of HPC-X/C*n* were identified using both ATR FT-IR and ^1^H-NMR spectral measurements. Here, the ATR FT-IR and ^1^H-NMR spectra of pristine HPC, HPC-Pr/C*7*, HPC-Bu/C7, and HPC-Pe/C3 are shown as the typical examples (Appendix A).

When ATR FT-IR spectra of pristine HPC and esterified HPC derivatives were compared, a broad peak of the O-H stretching vibration around 3100–3600 cm^−1^ thoroughly disappeared after the esterification of pristine HPC. Furthermore, a sharp peak of C=O stretching vibration concomitantly appeared at 1700 cm^−1^ (Appendix A). Such a change in the FT-IR spectrum suggests that the hydroxy groups of HPC were completely substituted with alkanoyl groups by the esterification [10].

Subsequently, ^1^H-NMR spectra were measured to quantitatively determine the X*E* and *CnE* values, defined in the Experimental Section, by the integrated values of the peaks (Appendix A). The detailed analysis procedures and assignment of the peaks in ^1^H-NMR spectra are described in the Appendix A. Both the *XE* and *CnE* values of HPC derivatives are summarized in Table 1, Table 2 and Table 3. The *XE* and *CnE* values were ~2.7 and ~0.3, respectively. These values suggest that the mixed ratio of two kinds of alkanoyl groups can be controlled at approximately 9:1, and that hydroxy groups in HPC are completely substituted with alkanoyl groups because of the total *XE* and *CnE* values of ~3.0.

The complete substitution is quite important for a fair comparison of reflection properties of HPC derivatives. This is because the substitution degree of HPC has a great impact on the reflection characteristics. For instance, according to the previous report on HPC-Bu with different *BuE*, their reflection peak wavelength red-shifted from 401 nm to 835 nm as *BuE* decreased from 2.96 to 2.20 [36]. In this study, the *XE* and *CnE* values were standardized at ~2.7 and ~0.3, respectively, to elucidate the pure influence of the side-chain length on reflection properties. These values mean that hydroxy groups in pristine HPC were completely substituted with alkanoyl groups. It should be noted that the *PeE* values of some HPC-Pe/C*n* were ~2.4 due to the low esterification reactivity of pentanoyl chloride, probably due to relatively long alkyl chains. 

TGA measurements also confirmed that the HPC derivatives did not degrade during the heating process up to 350 °C. As evidenced from the TGA curves of pristine HPC and HPC-Pr, their decomposition occurred above 200 °C (Appendix A). This temperature was much higher than the temperature range to measure the reflection properties of HPC derivatives, as described in Section 3.2, Section 3.3 and Section 3.4. It should be stressed that the thermal decomposition of HPC derivatives hardly occurs during their optical measurements.

### 3.2. Reflection Properties of HPC-X (X = -Pr, -Bu, and -Pe)

The reflection peak of HPC-X (X = -Pr, -Bu, and -Pe) shifted to a longer wavelength as the temperature increased. The thermally induced red-shift of reflection peak wavelength is consistent with a previous study [21]. Figure 1A shows the changes in the transmission spectrum of HPC-Pe as a function of temperature on the heating process. Although HPC-Pe exhibited a reflection peak at 611 nm, observed as a red reflection color, at 30 °C, the heating treatment resulted in a continuous red-shift of the reflection peak to the infrared wavelength region. For instance, when heated to 100 °C, the reflection peak reached 922 nm. Similarly, the reflection peak of HPC-Pr also red-shifted from 419 nm to 659 nm by heating from 100 °C to 140 °C (Appendix A), and HPC-Bu from 404 nm to 661 nm by heating from 40 °C to 90 °C (Appendix A). Such a red-shift in reflection peak wavelength during the heating process can be ascribed to an increase in the *p* value due to the thermotropic CLC property [21,37,38].

As the temperature of HPC-X approached the isotropic phase transition temperature (*T*_i_), the reflection peak gradually broadened and attenuated. This is because the helical orientation of CLCs was disturbed, and the distribution of the *p* value widened by thermodynamic molecular motion when the temperature of CLCs approached the *T*_i_. For instance, the reflection peak in the transmission spectrum of HPC-Pe at 100 °C was found to be broader and smaller than the peak at 30 °C (Figure 1A). This tendency was especially apparent for the transmission spectra of HPC-Pr (Appendix A), and can be ascribed to the relatively high *T*_i_ of HPC-Pr. The *T*_i_ of HPC-Pr is ~150 °C according to our previous study [39], in which the *T*_i_ of HPC-Pr was determined by polarized optical microscopic observation under crossed-Nicol upon the heating process. The *T*_i_ of HPC-Pr was the highest among the three kinds of esterified HPC derivatives with different alkyl chains of alkanoyl groups: HPC-Pr, HPC-Bu, and HPC-Pe [21,33]. It was anticipated that the effect of thermal motion of the molecules around *T*_i_ on the reflection property was the largest, resulting in the broadening of the reflection peak of HPC-Pr near its *T*_i_.

The wavelength shifts of the reflection peaks of HPC-X (X = -Pr, -Bu, and -Pe), which were dependent on the temperature due to the thermotropic CLCs, were fully reversible unless the temperature did not exceed the *T*_i_ of HPC-X. It should be noted that even if the temperature exceeded *T*_i_ once, the reflection peak could be revived by applying mechanical stimuli, such as shear strain [40]. The wavelength shifts in the reflection peaks of HPC-Pr, HPC-Bu, and HPC-Pe upon the heating process, are summarized in Figure 1B. Thus, the reflection peak wavelengths and colors of HPC-X could be controlled by changing the temperature. 

According to a previous study [21] and the results shown in Figure 1B, it was confirmed that CLCs with reflection peaks below ~600 nm, that is, blue and green reflection colors, at room temperature could not be fabricated when HPC derivatives with a single component of alkanoyl group in their side chains were used. Both HPC-Pr and HPC-Bu were expected to exhibit a reflection peak in ultraviolet wavelength region below 400 nm at room temperature. If the chain length of the alkanoyl group was further extended than the pentanoyl group, that is, an HPC derivative with a hexanoyl group, the reflection peak has been observed in the infrared wavelength region over 2000 nm at 20 °C [21]. Thus, the reflection peak of HPC-X tends to red-shift as the side chain length increases. This red-shift stemmed from an increase in the *d* value as well as a decrease in the *φ* value, which result in an increase in both the *p* and *λ* values, according to Equations (1) and (2). A previous study revealed that an introduction of a longer side chain leads to an increase in the *d* value and a decrease in the *φ* value [33]. This result implies that the long side chain induces consequent weakening of the chiral intermolecular interaction, which induces the molecular helical structure. Considering these results, it is not possible to continuously control the reflection peak wavelength and color of HPC-X at room temperature. Thus, CLCs with visible reflection at room temperature cannot be produced by simply introducing one kind of alkanoyl group to HPC.

### 3.3. Side-Chain Length Effect of HPC-Pr/Cn, -Bu/Cn, and -Pe/Cn on Reflection Properties

When comparing the wavelength shift behavior of the reflection peak of HPC-Pr/C*n* by changing the temperature, we found that the reflection peaks emerged at longer wavelengths at the same temperature when the HPC derivatives had long alkanoyl groups in the side chains, corresponding to an increase in the *n* value. For example, the reflection peak of HPC-Pr/C*n* (*n* = 2, 3, 4, 5, 6, 7, 9, 11, and 13) at 95 °C increased by ~23 nm as *n* increased by one (Figure 2A).

This trend was no longer valid at temperatures above 110 °C: for instance, the reflection peak of HPC-Pr/C5 exceeded that of HPC-Pr/C6. This can be ascribed to the broadening of the reflection peak when approaching the temperature near *T*_i_ (Appendix A). The same behaviors of reflection wavelength were also observed in HPC-Bu/C*n* (Appendix A) and HPC-Pe/C*n* (Appendix A).

The region of temperature range at which HPC-Pr/C*n* (*n* = 2, 3, 4, 5, 6, 7, 9, 11, and 13) exhibited a reflection peak in a visible wavelength range shifted toward lower temperatures (Figure 2A), but that of the temperature range of HPC-Pr/C*n* (*n* = 13, 15, and 17) with a reflection peak in a visible wavelength range shifted toward higher temperatures (Figure 2B). The mechanism of these shifts in the temperature range will be discussed in the following section.

To elucidate the relationship between side chain length and reflection wavelength in more detail, Figure 3A plots the temperature of HPC-Pr/C*n* with a reflection peak at 500 nm. The reflection peak wavelength of 500 nm was chosen because the temperature of the reflection peak of 500 nm was far away from *T*_i_. The CLC orientation is not considered to be disturbed, allowing for a fair comparison of the effects of side chains on the reflection properties. The temperature was lowered from 113 °C to 74 °C as *n* increased from 2 to 13 (Figure 3A, blue arrow). This temperature lowering was caused by the enlargement of the *p* value. This is reasonable because the increase in the side chain length contributed to the increase in *p* value and the red-shift of the reflection peak wavelength, as described in Section 3.2. On the other hand, the temperature was raised from 74 °C to 85 °C by increasing the *n* value from 13 to 17 (Figure 3A, red arrow). This increase probably occurred from the enhancement in van der Waals interactions. When the alkyl chain length was extended even longer, van der Waals interactions among side chains were augmented, and intermolecular interactions intensified, resulting in a decrease in *d* value. As a result, the temperature required to reflect light at 500 nm began to increase at *n* = 13.

The same tendency was also observed for HPC-Bu/C*n* (Figure 3B). The temperature at which HPC-Bu/C*n* showed a reflection peak at 600 nm lowered from 85 °C to 34 °C as the *n* increased from 2 to 11 (Figure 3B, blue arrow), and then increased to 43 °C at *n* = 17 (Figure 3B, red arrow). Note that a plot at *n* = 13 is omitted because a reflection peak of HPC-Bu/C13 was not observed in the visible wavelength range. Although a very broad reflection peak at ~670 nm appeared in the transmission spectrum of HPC-Bu/C13 at 30 °C, no reflection peak appeared at both 40 °C and 50 °C (Appendix A). Such broadening of the peak was also observed for HPC-Pr near its *T*_i_; it can be estimated that the *T*_i_ of HPC-Bu/C13 was probably ~40 °C, which prohibited the emergence of visible reflection at room temperature.

In the case of HPC-Pe/C*n*, a fair comparison of the temperature at which HPC-Pe/C*n* reflected a light of a certain wavelength was difficult because the *PeE* values were not standardized at ~2.7. Since the *PeE* value has a great impact on the reflection peak wavelength, as described in Section 3.1, the HPC-Pe/C*n* with different *PeE* values are not suitable to survey the effect of the side chain length on reflection properties. The temperature of HPC-Pe/C*n* with a reflection peak at 800 nm is summarized in Appendix A. The reflection peak of 800 nm was determined because HPC-Pe/C6 showed a reflection peak over 800 nm above room temperature.

### 3.4. Side-Chain Length Effect of HPC-X/C2 on Reflection Properties

The temperature dependences of reflection peak wavelengths of HPC derivatives possessing propionyl groups (HPC-Pr, HPC-Bu/C2, and HPC-Pe/C2) are shown in Figure 4. The reflection peak of HPC-Bu/C2 red-shifted from 415 nm to 671 nm when heated from 50 °C to 100 °C. Similarly, HPC-Pe/C2 showed the wavelength shift in reflection peak from 568 nm to 872 nm when heated from 30 °C to 100 °C. The temperatures at which the reflection peak reached 650 nm were ~136 °C for HPC-Pr, ~95 °C for HPC-Bu/C2, and ~51 °C for HPC-Pe/C2. The temperatures at which the same reflection wavelength stepwise decreased by ~40 °C as the number of carbons in the main component of alkanoyl group increased by one. This is based on our experimental results that the temperature dependence of reflection peak wavelength of HPC-X/C2 is almost the same. In other words, since the slopes of profiles in Figure 4 are almost ~5 nm/°C for all the esterified HPC derivatives, the above value of 40 °C is expected to remain almost the same for any reflection peak wavelength.

The difference of only one carbon number in the main alkanoyl group of esterified HPC derivatives had a significant influence on the reflection peak wavelength and the temperature range of visible light reflection. In the case of HPC-Pr/C*n*, when the *n* increased by one, that is, when the chain length of the minor alkanoyl group increased by one, the temperature at which HPC-Pr/C*n* reflected light at 500 nm decreased by ~5 °C in the *n* range between 2 and 4, as shown in Figure 3A. On the other hand, the temperature of HPC-X/C2 at which the same reflection wavelength decreased by ~40 °C as the number of carbons in the main-alkanoyl group increased by one, as shown in Figure 4. This experimental result implies that the structural change in the chain length of the alkanoyl group with a larger proportion of introduction has a more significant impact on the reflection properties of thermotropic CLCs of esterified HPC derivatives.

### 3.5. Room-Temperature Thermotropic CLCs of HPC Derivatives with Visible Reflection

Utilizing the above-mentioned reflection properties of the esterified HPC derivatives, it is possible to determine the conditions for preparing HPC derivatives with arbitrary reflection colors at room temperature. 

For instance, HPC-Bu/C6, HPC-Bu/C7, HPC-Pe/C2, and HPC-Pe were available because their reflection peaks are covered within the full visible wavelength range from 492 nm to 684 nm, even at room temperature. The reflection spectra of these HPC derivatives at 25 °C are shown in Figure 5. The light reflection intensities and wavelengths of reflection peaks of HPC-Bu/C6, HPC-Bu/C7, HPC-Pe/C2, and HPC-Pe were found to be 5.5% at 492 nm, 3.9% at 523 nm, 10.3% at 619 nm, and 7.9% at 684 nm, respectively. The reflection colors of HPC-Bu/C6, HPC-Bu/C7, and HPC-Pe/C2, were observed as blue, green, and red, respectively, as seen in the inset of Figure 5. The reflection image of HPC-Pe showed as dark red or invisible, due to its reflection peak appearing in the near infrared wavelength of 684 nm. 

By changing not only the length of alkanoyl groups but also the ratio of the two alkanoyl groups introduced, the reflection peak wavelengths of HPC derivatives at room temperature could be seamlessly tuned. We evaluated the temperature dependences of the HPC derivatives possessing propionyl groups, butyryl groups, or mixed propionyl and butyryl groups (Appendix A). The temperature range in which HPC-Pr reflected visible light was the highest, and the range shifted to the lower temperature side as the introduction ratio of butyryl group increased. These results mean that the reflection peak wavelength of an HPC derivative at a certain temperature can be modulated. Concretely, a change in the introduction ratio of two kinds of alkanoyl groups by 0.3 resulted in a spectral change in the reflection wavelength with ~44 nm. A similar result was also reported in a previous study [35]. In this way, by controlling the introduction ratio of the two kinds of alkanoyl groups, CLCs of HPC derivatives that reflect light at a desired wavelength at room temperature can be prepared. These CLCs are easy to handle because their reflection colors appear at room temperature without the aid of any solvents such as lyotropic CLCs, which can easily evaporate and critically affect the reflection properties of CLCs. Solvent-free CLC materials with reflection colors at room temperature are very attractive from the technological standpoint when the CLCs are used in functional devices such as external stimuli-responsive sensors, since they show stable reflection colors and are resistant to discoloration. Our findings would expand the possibilities for the technological applications of CLC materials to sustainable photonic devices using cellulose.

## 4. Conclusions

In this study, we successfully obtained CLCs that exhibit reflection colors at room temperature, without any solvents, by introducing two different lengths of alkanoyl groups into HPC. Furthermore, we found an interesting phenomenon, that the introduction of a long alkanoyl group as part of the side chain changes the temperature-dependent trend of the reflection property. This phenomenon is likely due to increased intermolecular interactions between alkanoyl side chains. Our findings pave the way for the preparation of room-temperature thermotropic CLCs of esterified HPC derivatives with visible reflection. This study can contribute to the development of versatile applications of CLCs utilizing cellulose for the realization of a sustainable society.

## Data Availability

Data are contained within the article.

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
