# Peer review of "Room-Temperature Cholesteric Liquid Crystals of Cellulose Derivatives with Visible Reflection"

_polymers, 2022, doi:10.3390/polym15010168_

Round 1
Reviewer 1 Report
The work is well written, easy to read and very interesting. However, in the last part, when they carry out the study at room temperature, there are experimental procedures that are not described and that are important in relation to the conclusions reached by the authors.
- HPC, what characteristics does it have, size?
- Pag. 4, line 138 “As esterifying remaining hydroxy groups of HPC with another alkanoyl group, propionyl chloride (C2H5COCl; 2.41 g, 1.06 g/mL, 26.0 mmol, 5.10 eq. to the number of anhydroglucose units in 2.00 g of HPC) was added dropwise. After reacting for additional 20 hours, the reaction solution was purified by dropping the solution into 500 mL of ultrapure water. The reaction conditions are the same in the second esterification?
- Pag. 5, line 170: “Finally, the HPC derivatives were sandwiched between a pair of the substrates with a gap of ~200 μm on a hot stage at 100−120 °C” In this cell preparation step, what concentration of HPC do you use, do you use aqueous suspensions? This part should be more detailed.
- Pag. 6, line 220: “Such a red-shift of reflection peak wavelength in the heating process can be ascribed to the increase in the p value due to the thermotropic CLC property”. Could you add some reference that confirms this statement?
- 3.5. Room-Temperature Thermotropic CLCs of HPC Derivatives with Visible Reflection. Figure 5: The insets denote the reflection images of HPC derivatives at room. The authors refer to HPC derivatives, are they films? how they were prepared is not described in materials and methods, and this is crucial to be able to make comparisons.
Author Response
Manuscript ID: polymers-2093326
Manuscript Title: Room-Temperature Cholesteric Liquid Crystals of Cellulose Derivatives with Visible Reflection
Authors: Yuki Ogiwara, Tatsuya Suzuki, Naoto Iwata, and Seiichi Furumi*
Point-by-point responses. Our responses and opinions are shown in BLUE.
Reviewer #1
============================================================
To Reviewer #1
Thank you very much for your kind review and generous encouragement to our report. Our manuscript has been revised according to your comments.
============================================================
Comment #1
HPC, what characteristics does it have, size?
Response #1
In this study, we used HPC with complete solubility in acetone at appropriate concentration, but not cellulose nanocrystals. After esterification of HPC with alkanoyl chlorides in acetone, the products are sticky polymer melts of thermotropic cholesteric liquid crystals with visible reflection.
The characteristics of HPC are described in Lines 126–136 of Pages 3–4. The weight average molecular weight and number average molecular weight of HPC were determined to be 4.45×104 and 2.30×104, respectively, by measurement with size-exclusion chromatography calibrated using polystyrene standards.
Comment #2
Pag. 4, line 138 “As esterifying remaining hydroxy groups of HPC with another alkanoyl group, propionyl chloride (C2H5COCl; 2.41 g, 1.06 g/mL, 26.0 mmol, 5.10 eq. to the number of anhydroglucose units in 2.00 g of HPC) was added dropwise. After reacting for additional 20 hours, the reaction solution was purified by dropping the solution into 500 mL of ultrapure water. The reaction conditions are the same in the second esterification?
Response #2
The reaction conditions of the second esterification are the same as the first one except for the reaction time. In response to your opinion, we revised the following sentence in Lines 172–176 of Page 5:
As esterifying remaining hydroxy groups of HPC with another alkanoyl group, propionyl chloride (C2H5COCl; 2.41 g, 1.06 g/mL, 26.0 mmol, 5.10 eq. to the number of anhydroglucose units in 2.00 g of HPC) was added dropwise into the reaction solution under continuous stirring and heating at the same temperature.
Comment #3
Pag. 5, line 170: “Finally, the HPC derivatives were sandwiched between a pair of the substrates with a gap of ~200 μm on a hot stage at 100−120 °C” In this cell preparation step, what concentration of HPC do you use, do you use aqueous suspensions? This part should be more detailed.
Response #3
We did not add any solvents or chiral dopants to the esterified HPC derivatives as they can show thermotropic CLC phase and reflection colors in the absence of such additives. In the case of CNCs, CLC phase can be observed when they are dispersed in water. Similarly, pristine HPC can also show CLC phase when it was dissolved in solvents such as water. However, the esterified HPC derivatives can intrinsically show thermotropic CLC phase in the absence of additives such as solvents or chiral dopants. This unique property of esterified HPC derivatives is attracting interests for decades since the pioneering works reported by Gray (Gray, D.G. et al., Int. J. Biol. Macromol. 1992, 14, 170–172 : Ref. 7) or Watanabe (Watanabe, J. et al., High Perform. Polym. 1999, 11, 41–48 : Ref. 21). Based on your comment, the following sentence was added in Lines 181–184 of Page 5 as follows:
All HPC derivatives synthesized in this study were obtained as sticky polymer melts. These HPC derivatives exhibited thermotropic CLC phase in the absence of any additives such as solvents or chiral dopants as will be discussed later.
In addition, the term of “as sticky polymer melts” was added in Lines 199–201 of Page 5 as follows.
The HPC derivatives as sticky polymer melts were sandwiched between a pair of glass substrates with a gap of ~200 μm on a hot stage at 100−120 °C.
Comment #4
Pag. 6, line 220: “Such a red-shift of reflection peak wavelength in the heating process can be ascribed to the increase in the p value due to the thermotropic CLC property”. Could you add some reference that confirms this statement?
Response #4
According to your comment, we cited Ref. 21, 37, and 38, at the end of this sentence, Line 264, Page 7.
Comment #5
3.5. Room-Temperature Thermotropic CLCs of HPC Derivatives with Visible Reflection. Figure 5: The insets denote the reflection images of HPC derivatives at room. The authors refer to HPC derivatives, are they films? how they were prepared is not described in materials and methods, and this is crucial to be able to make comparisons.
Response #5
We apologize for the lack of description of the method of measuring the reflection spectra. Since all HPC derivatives were obtained as sticky polymer melts, the reflection spectra were evaluated by sandwiching them in a pair of two glass substrates. Based on your comment, we added the description to the experimental section, Lines 210–213 of Page 6, as follows:
Reflection spectra of the CLC cells were recorded using the above-mentioned spectrometer and halogen light source. An optical fiber with a reflection probe was used for the measurements. All reflection spectra in this study were measured at room temperature of ~25 °C.
Furthermore, we emphasized that the HPC derivatives are not films and form polymer-melt, at Lines 204–206 on Page 6 as follows:
Importantly, the HPC derivatives are not solid-state films or powders but polymer-melts. The detailed results will be explained from Section 3.1.
Reviewer 2 Report
In this manuscript, the authors reported their research on room-temperature cholesteric liquid crystals of cellulose derivatives with visible reflection. The paper is interesting, but needs some revisions before its publication in Polymers.
1. Page 1 lines 37: Eliminate multiple references. Please check the manuscript thoroughly and eliminate all the lumps in the manuscript.
2. Page 1 lines 37-39: It is recommended to further elaborate the principles of CLC reflection color and Bragg reflection.
3. Eliminate the use of redundant words, e.g.as a result, however, that is, furthermore, finally. Revise all similar cases, as removing these terms would not significantly affect the meaning of the sentence.
4. It is suggested to add a highlight section to the article for the convenience of readers.
5. The author needs to provide the contributions of this study more specific.
6. Page 7 lines288-292: What is the potential reasons for the phenomena you mentioned, please explain it for details.
Author Response
Manuscript ID: polymers-2093326
Manuscript Title: Room-Temperature Cholesteric Liquid Crystals of Cellulose Derivatives with Visible Reflection
Authors: Yuki Ogiwara, Tatsuya Suzuki, Naoto Iwata, and Seiichi Furumi*
Point-by-point responses. Our responses and opinions are shown in BLUE.
Reviewer #2
============================================================
To Reviewer #2
Thank you very much for your kind review and highly encouraging comment. Our manuscript has been revised according to your comments.
============================================================
Comment #1
Page 1 lines 37: Eliminate multiple references. Please check the manuscript thoroughly and eliminate all the lumps in the manuscript.
Response #1
Thank you for your advice. Although we carefully checked this point, we confirmed that all references are original and our revised manuscript does not contain any multiple references. We also added some references according to the editor’s comment about self-citation.
Comment #2
Page 1 lines 37-39: It is recommended to further elaborate the principles of CLC reflection color and Bragg reflection.
Response #2
We added the following sentence in Lines 40–47 on Pages 1–2:
In the CLCs, the layers, in which rod-shaped liquid crystalline molecules are oriented, accumulate with slight changes in the azimuthal orientation vectors to form the helical molecular structure. The rod-like molecules of CLC exhibit anisotropy in the refractive index. Therefore, the CLC structure brings about the periodic modulation of refractive index, thereby resulting in the emergence of photonic band-gap(s) as photonic crystals in the photon dispersion diagram due to anomalous group velocity near the edge of the Brillouin zone [18].
Since the mechanism of the photonic band-gap or photonic effect in CLCs is outside the scope of this study, we referred to a review (Ref. 18) at Line 47 on Page 2, in which the mechanism is described in detail.
Comment #3
Eliminate the use of redundant words, e.g.as a result, however, that is, furthermore, finally. Revise all similar cases, as removing these terms would not significantly affect the meaning of the sentence.
Response #3
According to your suggestion, we have reduced the use of redundant words in our revised manuscript.
Comment #4
It is suggested to add a highlight section to the article for the convenience of readers.
Response #4
Based on your comment, we added supplemental descriptions about the scope and key results in this study for the convenience of readers, in Lines 437–440 on Pages 11–12, as follows:
The solvent-free CLC materials with reflection colors at room temperature are very attractive from the technological standpoint when the CLCs are used in functional devices such as the external stimuli-responsive sensors.
Comment #5
The author needs to provide the contributions of this study more specific.
Response #5
We apologize for the lack of contribution to conceptualization. We rewrote the contributions in Lines 463–467 on Page 12 as follows:
Y.O. conducted experiments and analyzed the data and prepared a draft of the manuscript. T.S. also conducted experiments. N.I. analyzed the data and revised the manuscript with Y.O. S.F. conceived and supervised this project, and prepared the final version of the manuscript. All authors have read and agreed to the published version of the manuscript.
Comment #6
Page 7 lines288-292: What is the potential reasons for the phenomena you mentioned, please explain it for details.
Response #6
The reason was described in detail in Lines 302–303, Page 8 and Lines 342–345 of Page 8. For the convenience of readers, a citation to this section was added in Lines 334–335 on Page 9 as follows:
The mechanism of these shifts in the temperature range will be discussed in this proceeding section.
Reviewer 3 Report
The manuscript is very well written; there are some modifications that can make the paper more interesting to reviewers.
[1] mention advantageous of HPC over CNCs in their liquid crystal formation ability
[2] mention Condition with which HPC be used with lyotropic CNCs...
[3] mention potential application of HPC CLC coloration
[4] authors have to provide TGA of HPC as it might degrade at elevated temperatures
[5] please enrich introduction with discussion on liquid crystal formation of cellulose, following references can be used:
- Shopsowitz, K. E., Hamad, W. Y., & MacLachlan, M. J. (2012). Flexible and iridescent chiral nematic mesoporous organosilica films. Journal of the American Chemical Society, 134(2), 867-870.
- Matsumoto, K., Ogiwara, Y., Iwata, N., & Furumi, S. (2022). Rheological Properties of Cholesteric Liquid Crystal with Visible Reflection from an Etherified Hydroxypropyl Cellulose Derivative. Polymers, 14(10), 2059.
- Chan, C. L. C., Lei, I. M., van de Kerkhof, G. T., Parker, R. M., Richards, K. D., Evans, R. C., ... & Vignolini, S. (2022). 3D printing of liquid crystalline hydroxypropyl cellulose—toward tunable and sustainable volumetric photonic structures. Advanced Functional Materials, 32(15), 2108566
Author Response
Manuscript ID: polymers-2093326
Manuscript Title: Room-Temperature Cholesteric Liquid Crystals of Cellulose Derivatives with Visible Reflection
Authors: Yuki Ogiwara, Tatsuya Suzuki, Naoto Iwata, and Seiichi Furumi*
Point-by-point responses. Our responses and opinions are shown in BLUE.
Reviewer #3
============================================================
To Reviewer #3
Thank you very much for your kind review and comment. Our manuscript has been revised according to your comments.
============================================================
Comment #1
mention advantageous of HPC over CNCs in their liquid crystal formation ability
Response #1
According to your comment, we added the advantage of thermotropic cholesteric liquid crystals of HPC derivatives over cholesteric liquid crystals of CNC in Lines 61–74 of Page 2:
Hydroxypropyl cellulose (HPC), which is one of the derivatives of cellulose, is well known to exhibit thermotropic CLC phase by chemically modifying the terminal hydroxy groups of HPC [12–14]. Cellulose nanocrystals are also known to possess the intrinsic capability to generate CLC phase. However, they can exhibit CLC phase only when they are dispersed in water at appropriate concentrations [8,11], corresponding to a sort of lyotropic CLC phase. Thus, the thermotropic CLCs of HPC derivatives are easy to handle rather than the lyotropic CLCs of cellulose nanocrystals. This is because the reflection properties of lyotropic CLCs of cellulose nanocrystals would be quite vulnerable under dry atmospheric conditions due to the concentration increase by natural evaporation of water as the dispersion medium. On the other hand, some kinds of HPC derivatives show sufficient stability of thermotropic CLC phase with visible reflection even under dry conditions, arising from the solvent-free system of thermotropic CLC materials. As the other technological advantage, the chemical modifications of HPC allow the on-demand control of reflection properties of thermotropic CLCs.
Comment #2
mention Condition with which HPC be used with lyotropic CNCs.120°C.
Response #2
We did not add any dispersants or solvents to esterified HPC derivatives because the HPC derivatives exhibit thermotropic rather than lyotropic CLC phase, and they show reflection colors without using any solvents. Based on your comment, we emphasized that the CLCs in this study are solvent-free by adding the description in Lines 181–184 of Page 5 as follows:
All HPC derivatives synthesized in this study were obtained as sticky polymer melts. These HPC derivatives exhibited thermotropic CLC phase in the absence of any additives such as solvents or chiral dopants as will be discussed later.
Comment #3
mention potential application of HPC CLC coloration
Response #3
We mentioned potential applications of the CLC coloration in Lines 57–60 of Page 2. To emphasize the usefulness of CLC coloration from cellulose, the following explanation was added in Lines 90–92 of Page 2:
Taking this perspective in account, the thermotropic CLCs of cellulose derivatives with visible reflection at room temperature are very attractive soft materials for the sustainable photonic device applications.
Comment #4
authors have to provide TGA of HPC as it might degrade at elevated temperatures
Response #4
Based on your comment, we newly measured TGA of HPC and HPC-Pr to check for the degradation at elevated temperatures. As a result, we confirmed that the CLCs were not degraded in the temperature range of this study. We have added the TGA results to Figure S3 in the Supplementary Materials. The measurement methods are available in Lines 190–193 of Page 5 as follows:
The degradation of the HPC derivatives at heating process was monitored by using a thermogravimetric analysis (TGA) system (TGA 2010SA, NETZSCH, Selb, Germany). The TGA measurements of pristine HPC and HPC-Pr were conducted between 25 °C and 350 °C at the heating rate of 10 °C/min under a nitrogen atmosphere.
Comment #5
please enrich introduction with discussion on liquid crystal formation of cellulose, following references can be used:
Shopsowitz, K. E., Hamad, W. Y., & MacLachlan, M. J. (2012). Flexible and iridescent chiral nematic mesoporous organosilica films. Journal of the American Chemical Society, 134(2), 867-870.
Matsumoto, K., Ogiwara, Y., Iwata, N., & Furumi, S. (2022). Rheological Properties of Cholesteric Liquid Crystal with Visible Reflection from an Etherified Hydroxypropyl Cellulose Derivative. Polymers, 14(10), 2059.
Chan, C. L. C., Lei, I. M., van de Kerkhof, G. T., Parker, R. M., Richards, K. D., Evans, R. C., ... & Vignolini, S. (2022). 3D printing of liquid crystalline hydroxypropyl cellulose—toward tunable and sustainable volumetric photonic structures. Advanced Functional Materials, 32(15), 2108566
Response #5
Thank you for your kind suggestions, and we added several references of Ref. 11 and 15 about liquid crystal formation of cellulose into the introduction in Line 38 of Page 1. We also added the discussion about CNC, which is one of the most well-known CLC from cellulose in Lines 63–65 of Page 2.
Round 2
Reviewer 1 Report
No comment